# *Aedes* Larva Detection Using Ensemble Learning to Prevent Dengue Endemic

**Md Shakhawat Hossain** [1,*], **Md Ezaz Raihan** [1], **Md Sakir Hossain** [1], **M. M. Mahbubul Syeed** [2], **Harunur Rashid** [3] **and Md Shaheed Reza** [3]

1 Computer Science Department, American International University-Bangladesh, Dhaka 1229, Bangladesh
2 Department of Computer Science, Independent University Bangladesh, Dhaka 1229, Bangladesh
3 Department of Fisheries Technology, Faculty of Fisheries, Bangladesh Agriculture University, Mymensingh 2202, Bangladesh
* Correspondence: mshimul86@gmail.com

**Abstract:** Dengue endemicity has become regular in recent times across the world. The numbers of cases and deaths have been alarmingly increasing over the years. In addition to this, there are no direct medications or vaccines to treat this viral infection. Thus, monitoring and controlling the carriers of this virus which are the *Aedes* mosquitoes become specially demanding to combat the endemicity, as killing all the mosquitoes regardless of their species would destroy ecosystems. The current approach requires collecting a larva sample from the hatching sites and, then, an expert entomologist manually examining it using a microscope in the laboratory to identify the *Aedes* vector. This is time-consuming, labor-intensive, subjective, and impractical. Several automated *Aedes* larvae detection systems have been proposed previously, but failed to achieve sufficient accuracy and reliability. We propose an automated system utilizing ensemble learning, which detects *Aedes* larvae effectively from a low-magnification image with an accuracy of over 99%. The proposed system outperformed all the previous methods with respect to accuracy. The practical usability of the system is also demonstrated.

**Keywords:** *Aedes* larva detection; dengue epidemic; ensemble learning; transfer learning; stacking models; U-net segmentation

## 1. Introduction

Dengue fever is caused by the dengue virus, which is a positive-stranded ribonucleic acid (RNA) virus of the Flaviviridae family. The dengue virus has four antigenically distinct, but genetically connected serotypes, which are DENV-1, DENV-2, DENV-3, and DENV-4. Dengue is a mosquito-borne disease and is now endemic in more than 100 countries worldwide, including Bangladesh, according to the World Health Organization (WHO), with approximately 96 million cases reported annually [1]. The female *Aedes* mosquito is the sole transmission vector of this virus, which causes dengue fever or dengue hemorrhagic fever. This blood-feeding mosquito is also responsible for transmitting the Zika virus (ZIKV) and the Chikungunya virus (CHIKV). According to the Global Burden of Disease Study (GBD), Asia accounts for 70% of the global burden of dengue-caused disease, and Bangladesh is among the most severely affected countries in Asia. The first case of dengue fever was reported in 1960 in Bangladesh [2]. Forty years later, in 2000, a total of 5555 cases and 93 deaths were reported, and a local species of *Aedes* was also identified by entomologists [3,4]. Subsequently, dengue was declared an epidemic in Bangladesh. Since then, *Aedes* breeding and subsequent dengue cases have been regularly monitored. However, the number of cases continues to increase every year despite all the efforts made to control the outbreak of dengue. From 2003 to 2012, a total of 12,229 cases and 46 deaths were reported [5]. In 2017, another 2769 cases were reported, and this climbed to 10148 cases in 2018. In 2019, the country experienced

its largest outbreak, with 101,354 cases and at least 266 deaths [6]. The incidence rate was alarmingly high in 2019 compared to the previous years. According to the WHO, Bangladesh is now among the high-risk countries in the Southeast Asia region [7]. In 2021, from January to September, 15,000 cases and at least 57 deaths were reported. Consequently, it is now considered crucial to estimate the mortality of people in Bangladesh due to dengue [6,8]. In addition, these reports only account for the cases and deaths due to the dengue virus; they do not account for the CHIKV and ZIKV caused by the same *Aedes* vector. CHIKV outbreaks have been reported in more than 60 countries [9] and the ZIKV has been reported in 87 countries [10]. Bangladesh has experienced three major outbreaks of the CHIKV so far, which were in 2017, 2011, and 2008 [11].Dengue, Zika, and Chikungunya have emerged as serious global public health problems, particularly in Asia, Africa, and America. Bangladesh certainly has a high risk for these diseases, as it contains one of the most popular habitats of the *Aedes* mosquito, which is the common vector of these diseases [12–14]. These studies suggest that the current approaches are failing to monitor and control the breeding of *Aedes* mosquitoes in Bangladesh. In this paper, we identify the limitations of existing the *Aedes* monitoring systems and propose an effective monitoring system, which utilizes computer vision and machine intelligence to identify *Aedes* larvae to prevent *Aedes*-borne diseases. However, in this study, we mainly focused on endemic dengue to find a feasible solution to combat it.

Dengue virus is transmitted to humans by the bite of a dengue-affected *Aedes* mosquito, and a single bite of the dengue-carrying *Aedes* could be sufficient to transmit the virus to the subject's body and infect him/her. *Aedes* mosquitoes are the only biological vector that can transmit the virus from a dengue-affected human or animal to a healthy person, as illustrated in Figure 1. This virus cannot be transmitted by interaction between humans. Female *Aedes* mosquitoes, principally *Aedes aegypti*, are the main vectors of Dengue Serotypes 1–4. Though *Aedes aegypti* mosquitoes are the principal vector, *Aedes albopictus*, also known as the Asian tiger mosquito, and other *Aedes* species can also transmit the virus to varying degrees. Both vectors, *Aedes aegypti* and *Aedes albopictus*, are present in Bangladesh [12–14]. *Aedes* mosquitoes prefer to lay eggs in clear and still water hidden from sunlight, unlike Anopheles and Culex, which prefer dark and turbid water. Flower pots, buckets, tires, cups, barrels, bowls, or similar containers are major hatching sites for *Aedes*.

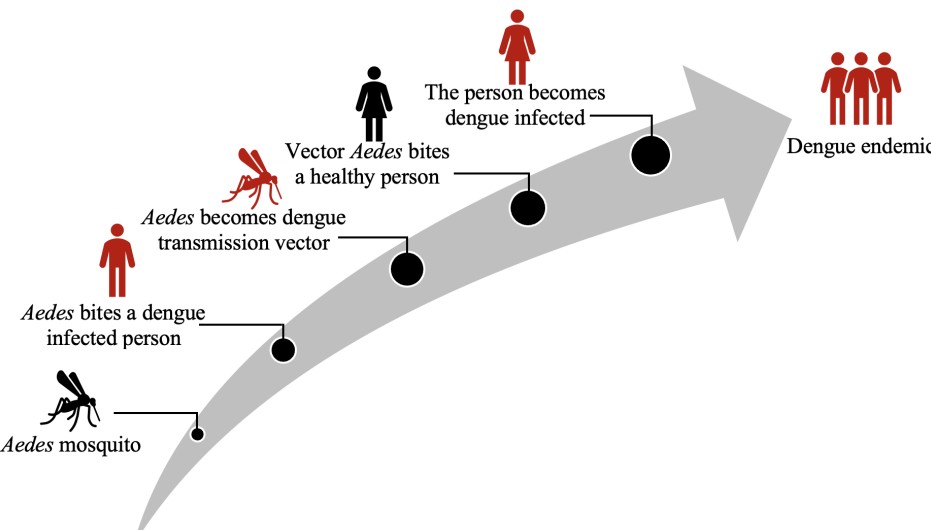

**Figure 1.** How the Aedes mosquito transmits the DENV into human.

The *Aedes* mosquito is holometabolous, and its lifecycle can be divided into four major stages of complete metamorphosis, which are egg, larva, pupa, and adult mosquito, as illustrated in Figure 2. Each stage differs significantly from the other in morphology, and it takes approximately 8 to 10 days for an adult mosquito to develop from an egg. *Aedes*

larvae feed on microorganisms in the water and can remain in this stage for days up to several months, depending on the habitat.

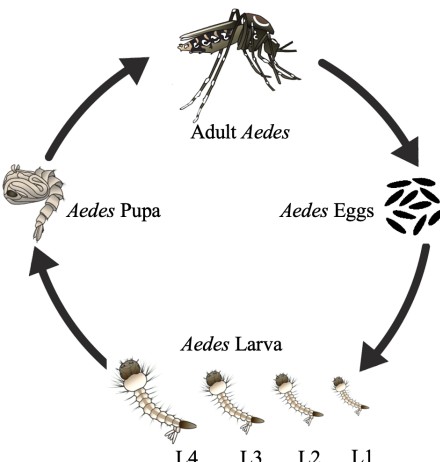

**Figure 2.** Life cycle of an *Aedes* mosquito from eggs to adult.

Usually, the larva develops into a pupa in 3 to 5 days during which they molt several times. After that, they don't molt anymore, therefore don't change their size at the pupa and adult stages. *Aedes* increases in size only at the larval stage which can be divided into four phases: 1st instar (L1), 2nd instar (L2), 3rd instar (L3) and 4th instar (L4). A larva could be 1/8 inch long at their 1st instar and grows to 1/2 inch at their 4th instar. From the larva stage, *Aedes* start becoming distinguishable from other species and can be identified by their head, neck, thorax and abdomen structure. The morphology of different abdominal parts such as mouth brush, palatum, preclypeal spines, mentum, compound eye, antenna, comb, siphon tube, pecten teeth anal papilla is useful to identify an *Aedes* larva [15–18].

Figure 3 shows *Aedes*, Culex and Anopheles mosquitoes at their pupa and larva stage which are the most common mosquito families found in Asia. *Aedes* larva is distinguished from the Anopheles and Culex utilizing different morphological features individually or in combination. Larva siphon, lies in it's abdomen is one such discriminating feature. Anopheles larva family has no siphon in the tails, and it is easiest among the three to identify at the larva stage. *Aedes* has a shorted and darker siphon compared to Culex. At the early instars of *Aedes* larva, the siphon remains soft which gets darker and harder in the later instars. The length and width of the *Aedes* siphon also increases with the larval growth, shown in Figure 4. The 4th instar siphon of *Aedes* larva is 0.79 mm long and 0.39 mm wide approximately which indicates that the siphon length is twice of its width. Another way to discriminate the *Aedes* larva from Culex is to observe their terminal segments. An *Aedes* larva has a comb-scale like silk pattern whereas a non-*Aedes* larva has an irregular pattern in comb, shown in Figure 5. In the laboratory, the entomologists mainly examine the comb-scale pattern to identify an *Aedes* larva. However, the examination of such features requires to use a microscope at a high magnification as 100× or more.

After the larva, they reach the pupa stage. Pupa also lives in the water and develops into an adult mosquito in 2 to 3 days. At this stage, *Aedes* and *Culex* show similar morphology, making it difficult to differentiate them. Once the pupa reaches to the adult stage, mosquitoes no longer stay in the water and fly away to live near people and bite them. Mosquitoes are easiest to identify in the adult stage. However, tracing and catching the mosquitoes for genera identification at the adult stage is not practical. Plus, it is not effective in preventing the dengue outbreak. On the other side, *Aedes* is very vulnerable at the larva phase and incapable of abandoning the site. Moreover, it cannot transmit any disease at this stage due to the lack of proboscis which they use to bite humans. Therefore, identifying and destroying the *Aedes* mosquito is most suitable at the larva stage to prevent dengue outbreak. The *Aedes* larva has distinguishable morphological features but this is

observable only through a microscope at a high magnification. Therefore, it is not possible to identify an *Aedes* larva with the naked eye. Additionally, it could be difficult to identify them using a microscope for an untrained person.

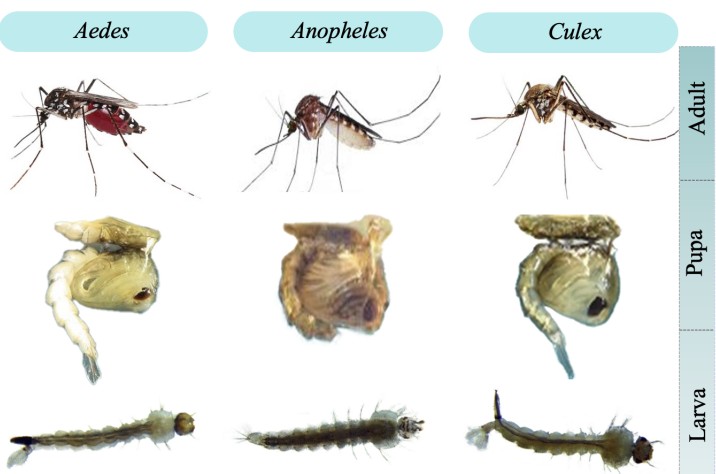

**Figure 3.** *Aedes*, *Anopheles*, and *Culex* mosquitoes at different stages of their life.

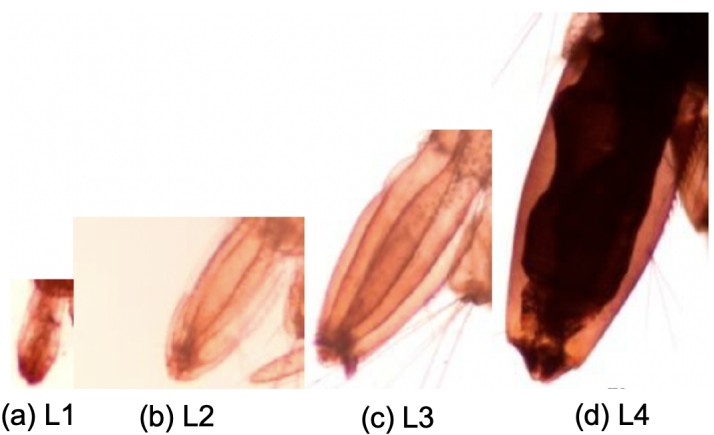

(a) L1          (b) L2          (c) L3          (d) L4

**Figure 4.** *Aedes* larva siphon at different instars at 108× magnification [19].

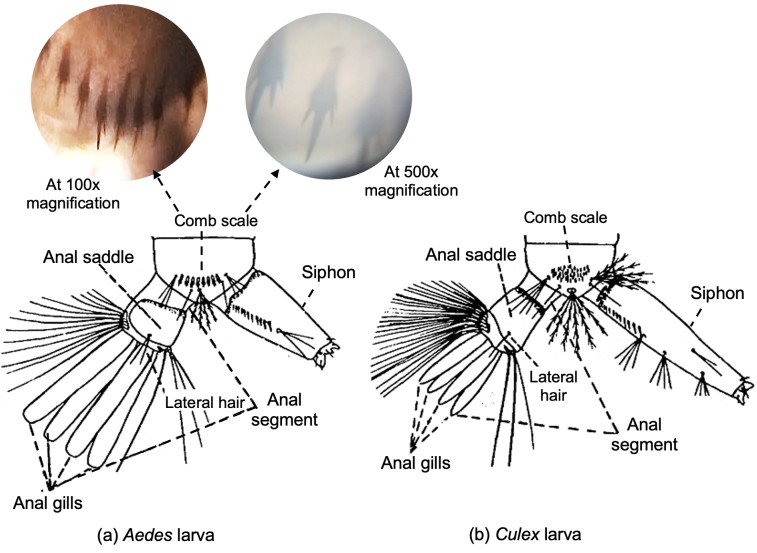

**Figure 5.** Higher order abdominal segmentation at larva stage.

The current *Aedes* monitoring system utilizes a light microscope equipped with a high magnification objective lens to identify the *Aedes* larva by an expert entomologist. The current practice of detecting the *Aedes* larva is a tedious process. Firstly, larva samples are collected from a mosquito hatching site. After that, it is transported to the laboratory which is usually at a considerable distance from the larva acquisition site. Then in the laboratory, an expert entomologist observes the larva using a microscope. The entomologists examine discriminating features under the microscope to identify an *Aedes* larva. This identification process is time-consuming, laborious, vulnerable to fatigue and can be done only by a trained entomologist. Usually, it takes a day to collect larva samples from the hatching site. Then, the observation under the microscope takes 10 min on average, plus a day at minimum to get the result from the lab. After that, the larva site is exterminated if the identification comes positive for *Aedes*. As this monitoring process is time-consuming, the larva develops into adult *Aedes* and flies away by the time steps are taken to destroy them. Another limitation is that the identification is subjective and the result could vary from person to person. Therefore, a more practical and effective system is required for identifying the *Aedes* larva to prevent the dengue endemic.

This paper presents an automated system which can identify *Aedes* larva from a digital image. The proposed system does not require to collect and carry the larva specimen to the laboratory for microscopic examination. More importantly, this system is not dependent on any expert's supervision and the detection result can be obtained in less than a minute without leaving the mosquito hatching site.

## 2. Related Works

The research conducted for identifying the *Aedes* mosquito can be broadly divided into three groups: ovitrap–gravitrap surveys, adult *Aedes* detection methods and *Aedes* larva detection methods. The ovitrap–gravitrap surveys are mainly intended to measure the distribution of *Aedes* mosquitoes by setting artificial breeding sites for adult mosquitoes where they are trapped and later lay eggs. This type of survey helps to estimate the possibility of a dengue outbreak. There are some methods developed to detect the adult *Aedes* [20–24]. However, in this work we focus on detecting *Aedes* larva to identify the *Aedes* larva hatching sites, so that the sites can be exterminated to prevent the dengue outbreaks. Destroying the *Aedes* at the larva stage is efficient and practical to combat the dengue endemic. A few solutions have been proposed for this purpose as well [25–30].

Sanchez-Ortiz et al. proposed a method to classify *Aedes* larva using a deep learning model [25,26]. This method utilized Alexnet, one of the early CNN models to classify the *Aedes* larva based on its comb pattern. This method requires a high magnification image of 100× and a good quality image to identify the pattern. The Alexnet was trained from scratch using 92 *Aedes* images and 198 non-*Aedes* images, which gave an accuracy of 96.8% for *Aedes* larva detection and 65% for the non-*Aedes* larva. The performance of deep learning models depends on the quality of training data and usually requires a large number of images to train the model from scratch, otherwise the generalized performance of the model could be poor. In a different work, Asmai et al. evaluated the performance of different CNN models such as VGG16, ResNet50, and InceptionV3 for the same task [27]. However, they utilized the transfer learning technique and trained only the final layers of the model using a limited number of *Aedes* larva images collected from different online sources. The VGG19 model achieved the highest accuracy of 87.26% for classifying the *Aedes* larva. Another larva classification method based on the comb pattern was proposed by Aris-ta-Jalife et al. [28]. They designed a low-cost image acquisition system by mounting a smartphone on a 60x microscope to capture the larva image on site. The images were transferred via a network that was then used for classification. The classification was performed by a VGG16 model that was pretrained on the COCO dataset and fine-tuned on 210 comb images of *Aedes* larva. As the image quality is important for identifying the comb pattern the quality of the image was evaluated before the classification. They also utilized a preprocessing step to segment the region of comb that was used for the classification.

The accuracy of segmentation and classification was 92.85% and 94.19%, respectively. All these methods utilize the same feature of comb pattern that requires a high magnification objective lens to observe. The comb pattern is the principal feature of these methods and it occupies only a small area of the image. Pre-processing can be utilized to focus only on the comb area before the classification. Moreover, the quality of the captured image should be high for detecting the patterns, which can be challenging to obtain at high magnification.

The method proposed by Azman et al. utilized the image of the entire larva, unlike the earlier methods that utilized only the comb segment of the body [29]. Azman et al. also designed a portable image acquisition system using a smartphone and microlens to capture the larva image. The captured image was then shared via a network for larva classification. A MobileNetV2 model was trained for larva classification and achieved an accuracy of 64.58%. This system takes approximately 10 min to obtain the final identification result. However, the detection accuracy is very low and the image magnification was not mentioned. However, the in situ image acquisition system proposed by them is useful for making a portable larva detection system that can be used by untrained people. De sliva et al. investigated the efficacy of digitally zoomed images of larva for automatic larva classification using CNN models [30]. In their study, the ResNet50 model was trained using larva images that were captured using a smartphone camera with $8\times$ digital zoom. The method resulted in 77.13% accuracy for larva classification using digitally zoomed images.

Existing automated larva detection solutions can be divided into two groups based on the imaging types: methods that use in lab high magnification microscopes and methods that rely on comparatively lower magnification portable imaging devices such as a smartphone or digital camera with magnifying gears. The acquisition of a high magnification image is a precondition for automatic larva detection and was achieved by the existing methods either using a high-magnification microscope or by magnifying the image of a low-magnification portable imaging device digitally or optically. The methods utilizing a high-magnification microscope tend to achieve higher accuracy compared to the methods that rely on portable imaging devices. The portable camera-based system had the highest accuracy of 77% which is not sufficient. The in-lab microscope-based system achieved the highest accuracy of 94%. However, the in-lab microscope-dependent automatic detection system requires the larva specimen to be transported from the hatching site to the lab, and, thus, failing to meet the time requirement to solve the problems of current manual evaluation. Although this approach eliminates the need for trained entomologists and reduces labor, it is not effective enough to prompt necessary control measures.

Therefore, we proposed an automatic larva detection system that can accurately identify the *Aedes* larva from a low-magnification image captured with a portable imaging device without needing to collect the specimen. The proposed system relies on artificial intelligence to detect *Aedes* larva without an expert's supervision and achieves a high accuracy that is comparable to the in-lab examination. This enabled the prompt implementation of appropriate control measures. Firstly, this method segments all the larva regions from the input image using U-net segmentation and eliminates the unnecessary background. After that, it classifies each segmented larva region as *Aedes* and non-*Aedes* using an ensemble model that was constructed by stacking optimally selected CNN models. The objective of this study was to develop an automatic larva detection method that can identify the *Aedes* larva from an image captured using a portable image device such as a smartphone or digital camera. The proposed method achieved an accuracy of 99% when demonstrated on low magnification images comparable to $2\times$ magnification that were captured using a digital camera. The results of the proposed method were compared with the existing automated larva detection methods to confirm its reliability. Lastly, this study also investigated the usability of the system for practical use.

## 3. Materials and Methods

### 3.1. Image Dataset

A dataset of 900 images was prepared from online sources and local entomologists for this study and includes 475 *Aedes* and 425 non-*Aedes* larva images. The non-*Aedes* set includes images of Culex and Anopheles larva. These images contained singular larva or multiple larva bodies. Among the 900 images, 67 images were manually annotated by an expert to create binary masks.

### 3.2. System Overview

The proposed system relies on a portable imaging device to capture images of larva hatching sites at low magnification. A digital camera equipped with a 65 mm f/2.8 1-5× microlens is used to capture the larva image at 1.4× magnification. Then, the system determines whether a larva in the captured image is an *Aedes* or non-*Aedes*. The framework of the proposed system is illustrated in Figure 6. The proposed system contains three major modules: image acquisition, larva body segmentation, and larva classification. The system starts with the image acquisition module that is responsible for capturing the input image, checking its quality, and for color normalization. Then in the segmentation module, each larva body is segregated from the input image to create a region of interest (ROI) for each larva. Subsequently, in each classification module every larva ROI is classified as *Aedes* or non-*Aedes*.

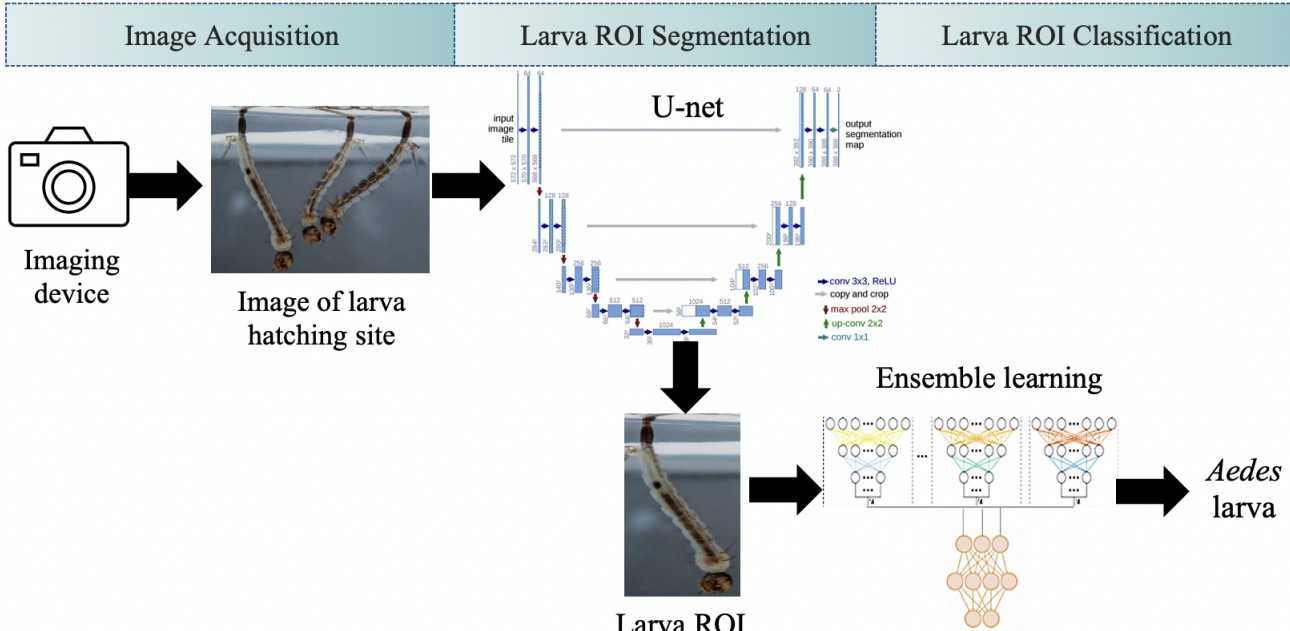

**Figure 6.** Structure of the proposed larva classification system.

Algorithm 1 explains the methodology of the proposed system. The algorithm starts by initializing the necessary parameters. Then, it estimates the quality index of the image to ensure that it is not suffering from focus blur [31,32]. For the quality evaluation, the gradient of the image is obtained using the Sobel filter. Then the difference between the local maxima and minima is calculated as the width of the edges. After that, the average width for the edges is calculated, which serves as the quality index. Blurry edges have small gradients that resulted in large width values compared to sharp edges. Therefore, we considered an image blurry and rejected larva identification if the quality was higher than 5.

---

**Algorithm 1** Aedes larva identification method

---

Initialization: $M \leftarrow \{m_1, m_2, m_3, ..., m_n\}$, $P \leftarrow \{P_1, P_2, P_3, ..., P_n\}$,
      $G = \{g_1, g_2, g_3, ..., g_n\}$, $f = \{f_1, f_2, f_3, ..., f_n\}$
**while** $I_{RGB}! = NIL$ **do**
    **if** $Quality_i \leq 5$ **then**
        **if** $C_{linear} \leq 0.0031$ **then**
            $I_{sRGB} = 12.92 \times C_{linear}$
        **else**
            $I_{sRGB} = 1.0552 \times C_{linear}^{\frac{1}{2.4}}$
        **end if**
        $I_R \leftarrow$ Resize $I_{sRGB}$ to $572 \times 572$
        Convert $I_R$ to segments, $S$ using U-net
        Apply a mask $m$, where $m \in S$, on $I_R$ to create $I_{ROI}$
        **while** $I_{ROI}! = NIL$ **do**
            $i = 1$
            **while** $i \leq n$ **do**
                $G_i \leftarrow$ Load base model $M_i$ with parameters $P_i$
                $f_i = G_i(I_{ROI})$
                $i = i + 1$
            **end while**
            Load meta model with the $n + 1$ parameters $\{\alpha, \beta, \gamma, \eta ... \lambda\}$
            $y = \alpha + \beta f_1 + \gamma f_2 + \delta f_3 + \eta f_4 + ... + \lambda f_n$
            **if** $y > 0.5$ **then**
                $\psi \leftarrow$ Aedes
            **else**
                $\psi \leftarrow$ Non-Aedes
            **end if**
        **end while**
    **end if**
**end while**
Output: $\psi$

---

Otherwise, the image is transformed to sRGB space to ensure the system is robust for different imaging devices. After that, the image is resized to $572 \times 572$ pixels for the U-net segmentation that extracts all larva ROIs from the resized image. The U-net takes 3 channel sRGB images as input and produces a binary mask as output, which is then post-processed to create a ROI for each segmented larva. After that, each ROI is fed into the classification module. We relied on an ensemble learning approach to achieve reliable classification and prepared a stacked model to identify the *Aedes* larva by combining the predictions from $n = 4$ optimally selected CNN models with the logistic regression classifier. The probability value of each CNN classifier is used in the logistic regression model to predict the class of the ROI. A larva ROI is classified as *Aedes* if the outcome of the regression model is greater than 0.5, otherwise it is classified as non-*Aedes*. The methods proposed previously encountered the over-fitting issue when training a single CNN model such as VGG16, VGG19, or ResNet models for larva classification which motivated us to utilize ensemble learning. The results of our experiment showed that the ensemble architecture achieves more reliable performance than individual models.

### 3.2.1. Larva ROI Segmentation

Segmentation of the region of interest is an important step that suppresses the background information to reduce the complexity of classification. The proposed system utilizes a U-net [33] network for segmenting the larva ROIs before the classification which improves the classification accuracy. Since the conventional U-net is a semantic segmentation, we annotated the input images to create binary masks for training the network. A set of 55 larva images were annotated manually for training the U-net network, as shown in

Figure 7. Let be $I_{RGB}$ is the set of captured images and $S$ is the set of possible larva body segments in image $I_R$ that we wish to extract. If $L$ is a particular larva segment of $I_R$, then $\mathcal{L}$ is represented as $\mathcal{L} = \phi(\hat{I})$ where $\phi$ is the labeling operator and $\hat{I}$ is a segmentation approximation of $I_R$. We trained the U-net based segmentation model $u_\theta : I_{RGB} \rightarrow S$ such that the segmentation of $I_R$ can be obtained as $\hat{I} = u_\theta(I_R)$ where $u_\theta$ is a non-linear function and $\theta$ is a vector of parameters. The parameter vector $\theta$ is derived from the training for which the accuracy of the segmentation model $u_\theta(I_R)$ is minimum. For the training, we prepared paired data as $(I_R, \hat{I})$ from the training image where $I_R$ is the input image to the segmentation model and $\hat{I}$ is it's segmentation ground truth. Then the vector $\theta$ is obtained during the training by minimizing the loss function which indicates the accuracy of the segmentation model, inversely. The loss function can be represented as the mean square error between the prediction and the plus a regularization term $(R_\theta)$ as given in Equation (1):

$$\hat{\theta} = arg \min_\theta \mathcal{L}(image, segment\_label, \theta)$$
$$= arg \min_\theta \mathcal{L} \sum_{I_{RGB}} \|u_\theta(I_R) - (\hat{I})\|^2 + \mathcal{R}_\theta(u_\theta(I_R)) \tag{1}$$

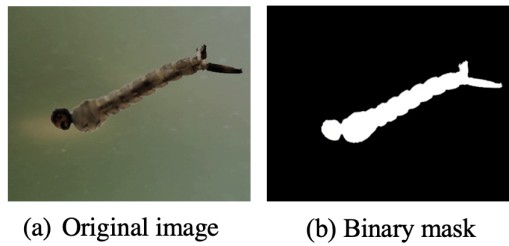

(a) Original image          (b) Binary mask

**Figure 7.** Binary mask for training the U-net network.

For training the U-net model, we used images that include both single and multiple larva bodies as the system is intended to detect *Aedes* larva from the image of a larva hatching site. Among the 55 images, 19 images contained a single larva body and the rest contained multiple bodies. Data was augmented by applying vertical flip, horizontal flip, and random zooming ($\times 1.0 \times 1.1$) for the training. The network was trained by an Adam optimizer with binary cross-entropy loss function and sigmoid output function. The epoch was 100 and the learning rate was 0.0001. The output of the U-net model is a binary image that is post-processed to create a ROI for each segmented larva.

3.2.2. Ensemble Learning for Larva Classification

The underlying principle of ensemble learning is inspired by the common saying "Together, we are stronger". Different machine learning models are trained differently on a dataset and have their limitations and strengths. Therefore, the errors made by different models are uncorrelated. Moreover, two models may achieve the same accuracy for a dataset, but one could be less efficient than the other. This indicates the trade-off while selecting a single best model among multiple models for a classification task. Instead of selecting a single model, ensemble learning provides an alternative solution to combine multiple models in a way that the strengths of multiple models are merged, and weaknesses are mitigated to achieve the best possible performance. Several studies have shown statistically and practically that an ensemble model can achieve significantly better predictive performance than a single model in different fields ranging from medical image analysis to agriculture [34–44]. Ensemble construction is computationally efficient as well. Training a single deep learning model often becomes computationally expensive if it is stuck on a local minimum during the optimization of the loss function. In an ensemble construction, the search is conducted from many different starting points which leads to the optimal point in a shorter time. Ensemble learning can be implemented in three ways: bagging, boosting, and stacking. We have utilized the stacking technique in which multiple

different models commonly known as base models are trained on the same dataset and then the prediction scores of base models are used to train another model called the meta-model to obtain the final result. The base models are trained on the training dataset while the meta-model is trained on the outputs for the base models to predict the final class label. Figure 8 illustrates the ensemble construction for the development of the larva classifier.

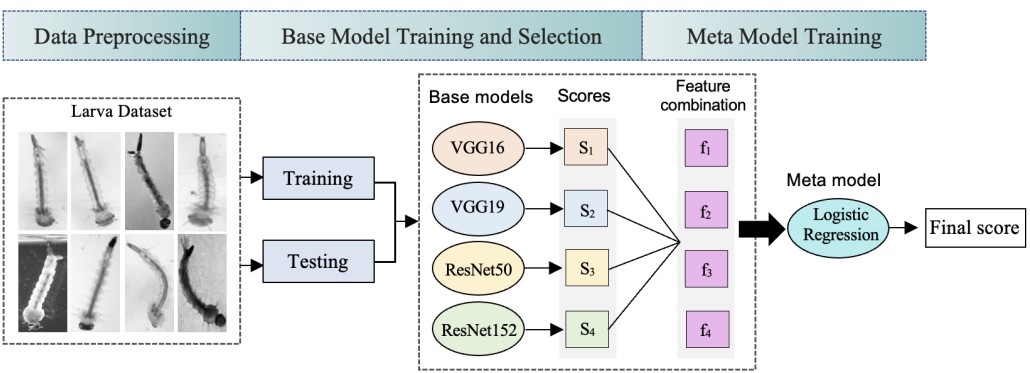

**Figure 8.** The overall process of constructing the larva classification model.

An individual machine learning classifier maps an input image to one of the predefined labels and can be defined as a $f_\phi : I_{image} \to C$ where $f_\phi$ is a function, $I_{image}$ is an image, and $C$ is the set of predefined classes. Unlike segmentation the classifier function uses a single label for each image. For the training, set of images and their associated class labels are provided. During training, the parameter vector $\phi$ is derived by optimizing the loss function of the network and it can be represented similarly according to Equation (2):

$$\hat{\varphi} = arg \min_\varphi \mathcal{L}(image, class\_label, \varphi)$$
$$= arg \min_\varphi \mathcal{L} \sum_{I_{ROI}} \|f_\varphi(I_{ROI}) - (\hat{I})\|^2 + R_\varphi(f_\varphi(I_{ROI})) \tag{2}$$

In this study, we trained 5 candidate models individually to predict the class label for a given image, which can be given as: $f_{VGG16} : I_{image} \to C$; $f_{VGG19} : I_{image} \to C$; $f_{ResNet50} : I_{image} \to C$; $f_{ResNet152} : I_{image} \to C$; and $f_{InceptionV3} : I_{image} \to C$. After that, four suitable models were selected among the candidates based on their performance. The process of selecting the base models is explained in Section 3.2.3. We have selected VGG16, VGG19, ResNet50, and ResNet152 as based models and rejected InceptionV3. Then a logistic regression classifier is used as the meta-model which is trained on the features derived from the prediction scores of base models. The outputs of the base models were used in the logistic regression model as the independent variable to predict the final class label as Equation (3):

$$y = ln\left(\frac{p}{1-p}\right)$$
$$= \alpha + \beta f_1 + \gamma f_2 + \delta f_3 + \eta f_4 \tag{3}$$

Here $p$ is the expected probability that the output is 1. $f_1$, $f_2$, $f_3$, and $f_4$ are the outputs of the base models $f_{VGG16}$, $f_{VGG19}$, $f_{ResNet50}$, and $f_{ResNet152}$, respectively, which serve as the independent predictors of the logistic regression model. $\beta$, $\gamma$, $\delta$, and $\eta$ are the parameter values commonly known as regression coefficients which were derived using maximum likelihood estimation. $\alpha$ is the intercept. The output of the logistic regression is 0 or 1 where 1 indicates the detection of an *Aedes* larva and 0 indicates the *Aedes* larva is absent.

The dataset used for training the base models is the larva ROIs extracted from the input images using U-net. In total 800 larva ROIs were used for training the base models. The logistic regression model is then trained on the 4-dimensional feature vectors generated for each input image to the base models. The output of the logistic regression is converted

to the class label as *Aedes* or non-*Aedes* larva. Another set of 100 ROIs unseen to the models was used for testing the proposed ensemble architecture in which it outperformed the existing methods in accuracy.

### 3.2.3. Base Model and Meta Model Selection

We relied on the deep learning approach and utilized the transfer learning technique for training the different CNN models as the candidate for base model selection. After that, the top 4 models were selected as the base model based on their performance. Table 1 shows the parameter values of the hyper-parameters investigated while training the base models. Figure 9 shows the training and validation curves for each candidate model using the best combination of parameters. Table 2 shows the loss and accuracy along with the area under the curve (AUC) for the candidates. Figure 10 shows the receiver operating characteristic (ROC) curves. Finally, Table 3 shows the comparison of candidates in terms of true positive rate or sensitivity (TPR), specificity or true negative rate(SPC) and accuracy (ACC) for a 4-fold cross-validation experiment.

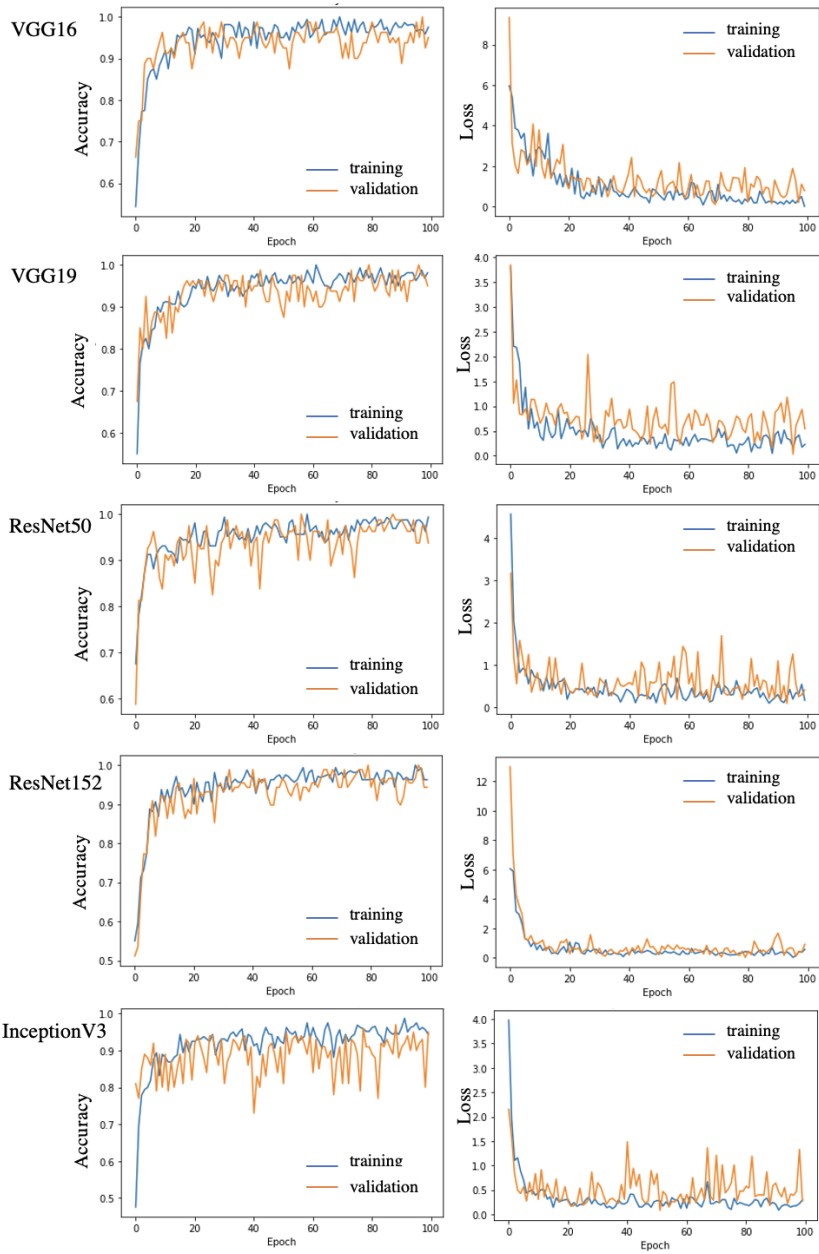

**Figure 9.** Model accuracy and loss for the models during training.

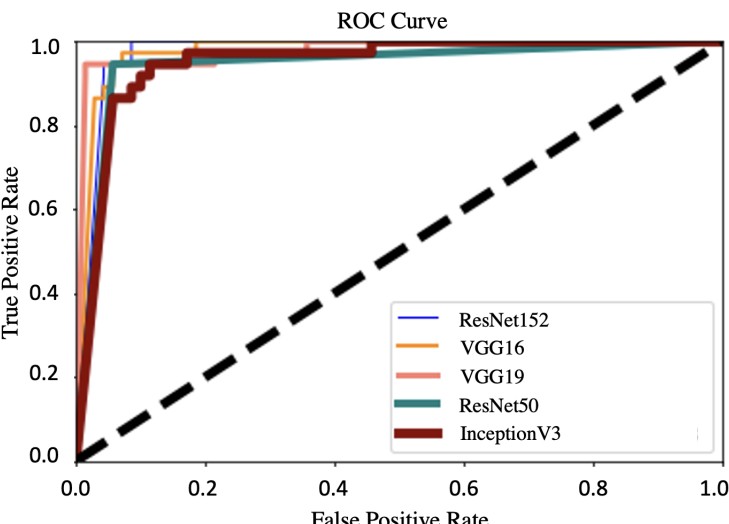

**Figure 10.** ROC curves of the candidates for the base model selection.

**Table 1.** Explored parameter values for the hyperparameters.

| Hyperparameters | Optimization Space |
|---|---|
| Epochs | [10, 30, 50, 70, 100] |
| Batch sizes | [5, 10, 20, 30] |
| Learning rates | [0.001, 0.01, 0.03, 0.05] |
| Dropouts | [0.5, 0.6, 0.7, 0.8] |

**Table 2.** Accuracy, loss, and AUC values for the candidate models.

| Model | Training Accuracy | Training Loss | Validation Accuracy | Validation Loss | AUC |
|---|---|---|---|---|---|
| VGG16 | 0.987 | 0.393 | 0.973 | 0.830 | 0.992 |
| VGG19 | 0.972 | 0.121 | 0.953 | 0.505 | 0.990 |
| ResNet50 | 0.975 | 0.172 | 0.920 | 0.160 | 0.978 |
| ResNet152 | 0.987 | 0.240 | 0.940 | 0.600 | 0.980 |
| InceptionV3 | 0.943 | 0.280 | 0.890 | 0.805 | 0.940 |

**Table 3.** Comparison of candidate models in 4-fold cross validation.

| | VGG16 | | | VGG19 | | | ResNet50 | | | ResNet152 | | | Inceptionv3 | | |
|---|---|---|---|---|---|---|---|---|---|---|---|---|---|---|---|
| | TPR | SPC | ACC | TPR | SPC | ACC | TPR | SPC | ACC | TPR | SPC | ACC | TPR | SPC | ACC |
| F1 | 0.94 | 0.95 | 0.97 | 0.95 | 0.95 | 0.97 | 0.94 | 1.00 | 0.95 | 0.91 | 1.00 | 0.92 | 0.92 | 0.95 | 0.88 |
| F2 | 0.93 | 1.00 | 0.98 | 0.95 | 1.00 | 0.97 | 0.97 | 1.00 | 0.98 | 0.91 | 1.00 | 0.92 | 0.82 | 0.94 | 0.88 |
| F3 | 0.90 | 0.93 | 0.94 | 1.00 | 0.97 | 0.96 | 1.00 | 0.97 | 0.97 | 0.97 | 0.91 | 0.97 | 0.80 | 1.00 | 0.95 |
| F4 | 0.98 | 0.94 | 0.92 | 0.98 | 0.91 | 0.93 | 0.86 | 0.97 | 0.96 | 0.92 | 0.94 | 0.94 | 0.92 | 0.90 | 0.84 |
| Avg. | 0.93 | 0.95 | 0.95 | 0.97 | 0.95 | 0.95 | 0.94 | 0.98 | 0.96 | 0.93 | 0.96 | 0.94 | 0.86 | 0.94 | 0.88 |

The deep learning method enabled fully automatic feature extraction and selection using CNN from images without human supervision. It has been shown that CNN achieves better performance for numerous applications compared to hand-crafted feature-based approaches such as SVM. However, it requires a large dataset for training. Due to the resource-hungry nature of CNN, models often become over-fitted when trained with a

limited dataset. Transfer learning is the careful tailoring of a trained CNN model to reuse it to solve a new problem. This is particularly important when enough labeled images are not available, which are a necessary component to training a CNN model from scratch [45,46]. In transfer learning, the weights of a previously trained model are transferred to another model to solve a new problem. Usually, the original model is trained on a large dataset. The underlying idea of transfer learning is inspired by the fact that the early layer features extracted by a CNN are identical across the domains, but the top layers of the network are specific to the details of the classes. Therefore, a model that is trained on a large dataset can be utilized to solve a new classification problem by freezing the early layers of CNN and adjusting the top layers according to the current classification task.

We have trained the VGG16, VGG19, ResNet50, ResNet152, and InceptionV3 candidates using transfer learning. The selection of candidate models was inspired by the previous research conducted in this field [21–26,28–30]. The candidates were pretrained on the ImageNet dataset, and we used our larva dataset for secondary training. We freeze the convolution base of the models and derived the bottleneck features for the larva dataset. The bottleneck features are the last activation maps before the fully connected layers of a model. Then, we trained a small fully connected model using the bottleneck features from the larva dataset on top of the convolution base. For training and validating a model, 800 images were used, which included 400 *Aedes* and 400 non-*Aedes* larva images. A total of 80% of the 800 images were used for training and 20% for validation. Another set of 100 images, which included 50 *Aedes* and 50 non-*Aedes* images, was used for testing. This experiment was performed for all the candidate models.

We used grid-based hyperparameter optimization to find the most suitable model architecture for the dataset. In gird-based optimization, the search space for each hyperparameter is discretized at first and then the search for the best architecture is conducted on a space of Cartesian product of all discretized hyper-parameters. The discretized values of selected hyper-parameters are shown in Table 1 for which a total of 384 ($6 \times 4 \times 4 \times 4$) combinations were tried during training. The grid-based searching suffers from the curse of dimensionality and the search time could be extremely high. Therefore, we selected an optimal discrete search space for the hyperparameters that not only saved time but also consumed fewer resources. For each network, the architecture that yields the best validation accuracy was selected as the candidate for base models. Later, the best 4 models were selected from the candidates from the K-fold cross-validation experiment to stack the models together for ensemble learning.

Data augmentation was also utilized during the training where the shear range, rotation range, and zoom range were 0.2, 0.2, and 15, respectively. Images were also augmented by flipping horizontally and vertically. The curves are shown in Figure 9 generated from the grid-search experiment. The accuracy, loss and AUC values are listed in Table 2. The high validation losses indicate the over-fitting issue of the candidates. After that, we further investigated the performance of the candidates in terms of sensitivity (true positive rate), specificity (true negative rate) and accuracy in 4-fold cross-validation. The result of this experiment is listed in Table 3. From Table 3 it can be observed that VGG16, VGG19, ResNet50, and ResNet152 perform better than the InceptionV3. Therefore, we have selected the VGG16, VGG19, ResNet50, and ResNet152 models as the base models. Then, these models were stacked to construct the ensemble network to combat the over-fitting issue.

We have also experimented with 4 candidates logistic regression (LR), support vector machine (SVM), K-nearest neighbors (KNN), and Naive Bayes classifier to select a suitable meta-model for the ensemble stacking. We estimated the stacked F-score, accuracy, and AUC value for the meta models. The LR model achieved the highest accuracy and stacked F-score of 0.99. The AUC value was 1.0 for the LR model. Therefore, we have selected the LR model as the meta classifier. Figure 11 shows the ROC curves for the different meta models.

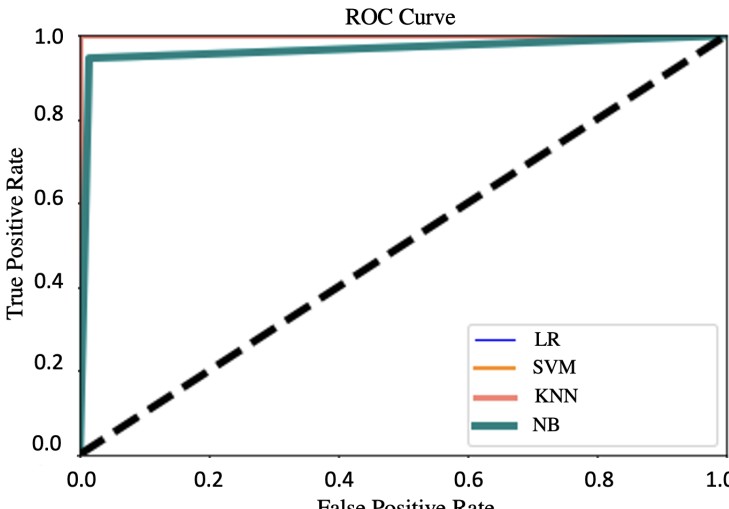

**Figure 11.** ROC curves for different meta models.

## 4. Results

*4.1. Evaluation of Larva Segmentation*

Firstly, we evaluated the performance of U-net based larva segmentation using the intersection over union (IoU) metric. The IoU measures the number of pixels common between the annotation mask prepared manually and the U-net prediction mask divided by the total number of pixels covered by both for an image, as given in Equation (4). Then, a true positive detection is detected if the IoU score exceeds a certain threshold. A false positive indicates the IoU score is less than the threshold, as illustrated in Figure 12.

$$IoU = \frac{Annotation \cap Prediction}{Annotation \cup Prediction} \tag{4}$$

**If threshold 0.8**

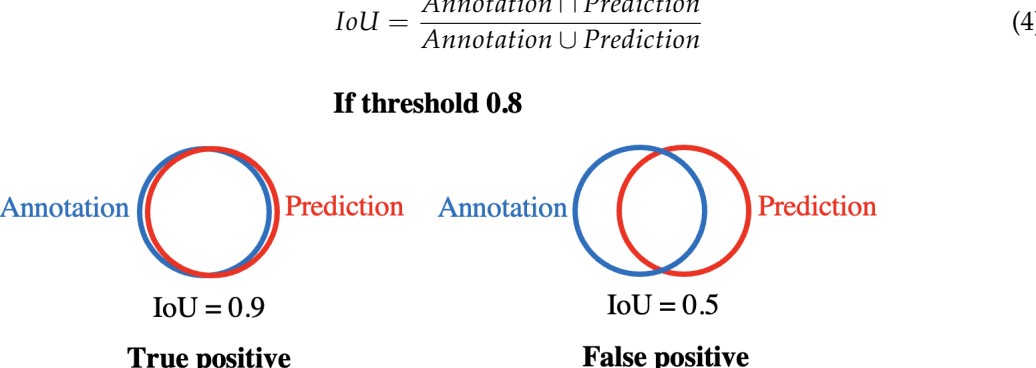

**Figure 12.** True positive and false positive estimation.

However, the IoU metric could be misleading when used to judge the performance of the segmentation model. For example, if the segmentation of a larva leaves off important body elements such as the siphon, even though it only takes up a small amount of space, the segmentation could be useless for the classification network. Therefore, we relied on the expert's manual evaluation to identify the false positives and the true positives. The U-net segmentation model was tested for 12 images that were unseen during the training. Among them, 4 images contained only one larva per image and the rest contained at least two larvae per image. The U-net model is capable of both detecting multiple larvae and single larva for a given input image. The average IoU score was 0.864 for the 12 images. The expert identified no false positives and false negatives for the 12 images. We also estimated the Dice coefficient, which was 0.843 the same as the IoU. Figure 13 shows the binary mask predictions made by the U-net model.

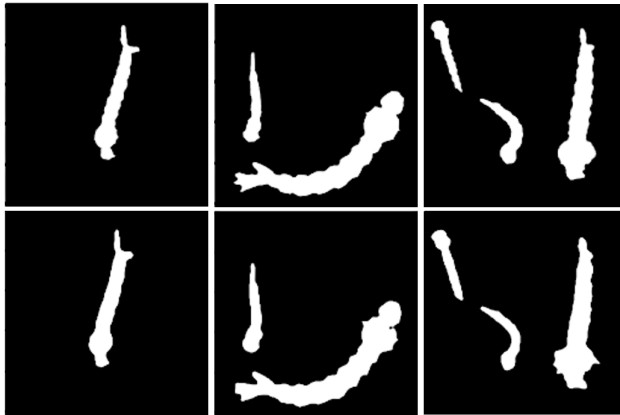

**Figure 13.** Expert's manual annotation masks (**top row**) vs. U-net predicted masks (**bottom row**).

### 4.2. Evaluation of Larva Classification

The proposed ensemble construction was trained using 800 larvae ROIs belonging to *Aedes*, Culex and Anopheles mosquitoes. Another set of 100 larvae ROIs was used for testing the proposed model. The proposed ensemble model was created by stacking four CNN base models VGG16, VGG19, ResNet50, and ResNet152. On top of that, logistic regression was used as the meta classifier to combine the results of the base models. The proposed ensemble model had an accuracy of 0.99, F-score of 0.99, specificity of 1, and sensitivity of 1 in the training. For the test dataset, the accuracy, F-score, specificity, and sensitivity were 0.98, 0.98, 0.98, and 0.97, respectively. From the results is it clear that the proposed stacking-based ensemble construction outperformed the classification performance of individual transfer learned models such as VGG16, VGG19, Res-Net50, ResNet152, and InceptionV3. The finding of this experiment satisfies the finding of earlier studies that stated that stacking multiple weak models improved the performance of individual models [31,33–42]. We have also compared the results of the proposed system with the earlier larva detection methods, as shown in Table 4. The proposed method outperforms the earlier methods in terms of accuracy. More importantly, it utilizes a comparatively lower magnification image. However, the methods were evaluated using different datasets.

**Table 4.** Comparison of proposed method with existing methods.

| Methods | Accuracy | Image Magnification |
|---|---|---|
| Sanchez-Ortiz et al. [25] | 96.8% | 100× |
| Asmai et al. [27] | 87.2% | - |
| Arista-Jalife et al. [28] | 94.1% | 60× |
| Azman et al. [29] | 68.5% | - |
| De sliva et al. [30] | 77.1% | 8× |
| Proposed method | 99% | 1.4×–4× |

### 4.3. Evaluation of Proposed System

The practical usability of the system was also demonstrated. Firstly, we identified the important parameters for designing a practical and effective larva identification system, which are time, image-magnification, identification type, identification accuracy, and cost. Then, we evaluated the feasibility of the proposed system considering these parameters.

This system does not require us to collect and carry the larva specimens to a lab. Additionally, it confirms the identity of the larva in less than 1 s, which significantly reduces the required time commitment. The proposed system spends the majority of its time (1 to 30 min) collecting an appropriate image and sending it over the network to the detection system that is installed on a computer. Despite this, the system takes far less time

overall than the conventional lab examination, which can take days to weeks. The proposed system relies on the computer vision and machine intelligence to identify the larva from a low-magnification images, such as $2\times$–$4\times$, with a 99% accuracy. This overcomes the need of a microscope and an expert's supervision. Another important issue for the practical implementation is the cost. This system utilize a digital camera with a microlens for image acquisition and a personal notebook (2.6 GHz Intel Core i5 Processor and 8GB RAM) for classification that has no GPU. However, it eliminates the cost of the microscope and the cost of delivering the specimen to a lab. Based on the above analysis, it can be asserted that the proposed system enables accurate and quick identification of *Aedes* larva for a practical and effective *Aedes* monitoring and controlling system.

## 5. Discussion

*Aedes* mosquito is the sole vector for transmitting dengue virus. Thus, the death caused by the dengue virus could be prevented if the hatching site of *Aedes* larva is monitored and necessary measures are taken to destroy the *Aedes* larva. However, it is important to ensure that only *Aedes* species are killed as the destruction of all the mosquitoes regardless their species could disrupt the ecosystem. For that reason, it is necessary to identify the *Aedes* mosquito for proper action. *Aedes* mosquitoes are most vulnerable at larva stage, and they leave the hatching site as soon as they turn to an Adult, making it difficult to trace and identity them. Plus, it is not effective to prevent dengue outbreak. Therefore, it is required to identify and destroy the *Aedes* mosquitoes at larva stage. Traditionally, the identification is performed by an expert entomologist using a microscope in the laboratory from the larva sample, collected from the hatching site. This process in time consuming, laborious, subjective and not practical.

Due to the advancement in information technology and artificial intelligence, it is now possible to capture images of larva hatching sites using a portable imaging device and classify the larva without human supervision. Several methods are proposed recently for that purpose, but they failed to achieve reliable accuracy and practical usability. In this paper, we have proposed a system that can identify the species of the mosquito from a larva hatching site image. Our system requires a comparatively low magnification image, and the larva identification accuracy is as high as 99%, which is significantly better than previous methods. Most of the past research have focused on the comb-scale pattern to identify an *Aedes* larva, which requires a high magnification image such as $100\times$ or higher. The proposed method uses the full larva body image at low magnification as $1.4\times$ to $4\times$.

The proposed system enables faster identification and prevention of *Aedes* mosquitoes by detecting the *Aedes* mosquito at larval stages. This system utilizes deep learning technology and requires a digital image of the larva hatching site. Firstly, it uses deep learning-based U-net segmentation to extract the region of interest, which is the larva body from the hatching site image. After that, an ensemble model is used for classifying each larva's body. The U-net based segmentation removes the unnecessary background information, which significantly reduces the complexity for the classifier and helps improve the performance of the system. The accuracy of the U-net segmentation was approximately 85%. This type of design, which uses segmentation as a form of preprocessing before classifying data, can be applied to numerous tasks, especially when it is needed to identify multiple objects from an image.

After segmentation, each larva body was classified using an ensemble model. This ensemble model combined multiple weak learners to utilize the strength of all the learners. In a primary investigation, we have tested the performance of individual classifiers using transfer learning. The accuracy of the individual classifiers was not satisfactory, and the model became over-fitted on many occasions. Previous studies have also reported similar findings. For that reason, we have constructed an ensemble model where the top four classifiers from the individual classifier studies were stacked to improve the performance and mitigate the weakness of each classifier. Finally, a logistic regression classifier was trained using the outputs of the stacked classifiers to predict the larva class. This proposed

framework has successfully identified the *Aedes* larva and outperformed the individual classifiers' results. Furthermore, this system outperformed all the previous methods in terms of accuracy.

The system was found to be practical and effective in the practical demonstration. However, we wish to further reduce the implementation cost in the future. This system requires a digital camera mounted with a microlens and a computer to run the proposed method. In the next step of our research, we plan to implement a system using a smartphone. This will make the system more practical, cheaper, more accessible, and easier to adapt. The proposed system will be more practical, more accurate, and will require less time and effort. Thus, it will play an important role in stopping the dengue endemic worldwide. The *Aedes* mosquito is also responsible for transmitting Zika and Chikungunya viruses. Hence, this system can also be utilized for combating Zika and Chikungunya disease with necessary modifications.

## 6. Conclusions

The proposed system can identify *Aedes* larva from a low-resolution image taken with a digital camera utilizing computer vision and deep learning technology. In the demonstration, it was found that the system is more accurate, quick, affordable, and useful. This kind of architecture can improve insect control and monitoring systems for mosquitoes or other insects, allowing timely action to be done.

**Author Contributions:** Conceptualization, M.M.M.S. and M.E.R.; methodology, M.M.M.S. and M.S.H. (Md Shakhawat Hossain); software, M.E.R.; validation, H.R., M.S.R. and M.M.M.S.; formal analysis, M.M.M.S.; investigation, M.M.M.S.; resources, H.R. and M.S.R.; data curation, M.M.M.S.; writing—original draft preparation, M.S.R.; writing—review and editing, M.S.H. (Md Shakhawat Hossain); visualization, M.S.H. (Md Sakir Hossain); supervision, M.M.M.S.; project administration, M.M.M.S.; funding acquisition, M.M.M.S. All authors have read and agreed to the published version of the manuscript.

**Funding:** This research received no external funding.

**Institutional Review Board Statement:** Not applicable.

**Informed Consent Statement:** Not applicable.

**Data Availability Statement:** The image data and python codes used in the experiment can be found at GitHub: https://github.com/Ezaz-Raihan/Aedes_Larva_Detection (accessed on 30 July 2022).

**Conflicts of Interest:** The authors declare no conflict of interest.

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
