# Peer review of "Aedes Larva Detection Using Ensemble Learning to Prevent Dengue Endemic"

_biomedinformatics, doi:10.3390/biomedinformatics2030026_

Round 1

Reviewer 1 Report

In this work, the authors proposed an ensemble based method for detecting Aedes larva from low magnification images and claimed to predict with an accuracy of 99%,

1. Introduction includes lot of text book information. This should be shortened.

2. The source for 900 images could be mentioned.

3. It is worth to discuss the selection of specific machine learning method to develop an ensemble based classifier.

4. Construction of test set should be briefed.

5. what is ? ? ? in page 372 and other places?

6. The data used for comparing other methods should be provided in the footnote.

Author Response

In this work, the authors proposed an ensemble-based method for detecting Aedes larva from low magnification images and claimed to predict with an accuracy of 99%.

Point 1: Introduction includes lot of textbook information. This should be shortened.

Response 1: Thank you very much for the comment. The manuscript has been revised and the explanation has been shortened. The following sentences are removed from the manuscript.

(From 2nd paragraph of page 2)

Aedes aegypti usually breeds inside while Aedes albopictus is an outside breeder. However, they can be found both indoors and outdoors. Both Aedes aegypti and Aedes albopictus don’t fly long distances, usually a few blocks. However, a single bite of a dengue-carrying Aedes could be sufficient to transmit the virus into the subject’s body and infect him.

(From 3rd paragraph of page 2)

An adult male Aedes lives 3 to 5 days whereas a female can survive as long as 2 months depending on the environment. The female Aedes is the one that bites humans and transmits the pathogens of dengue, chikungunya and zika viruses. An adult female Aedes lays eggs in a container above the waterline so that the eggs are submerged in the water for 2 to 3 days to hatch properly. A single female can lay 100 eggs at a time. The eggs depend on the water to hatch, and they turn into larva when covered with sufficient water.

Factors like temperature, salinity, pH value, nutrients and gas in the habitat determine the growth of larva. Extreme temperature, low nutrients and high salinity can lag the growth. Due to the dependency of the Aedes mosquito on the water, the rainwater or other unnatural sources of water that is trapped in some containers are the major habitats of the Aedes mosquito

(From 1st paragraph of page 4)

An adult Aedes is black and has white patches while Culex and Anopheles are yellowish in their adult phase. Adult mosquitoes can be identified simply by observing their resting position. Aedes} and Culex mosquito sit parallel to the surface while the Anopheles maintains an angle of 45 degrees approximately with the surface. Microscopic observation of the antenna can also reveal more discriminating morphology of an adult.

Point 2: The source for 900 images could be mentioned.

Response 2: The source of data is mentioned in the revised manuscript in section 3.1 Image Dataset as: “A dataset of 900 images was prepared from online sources and local entomologists for this study which includes Aedes and 425 non-Aedeslarva images.”

Point 3: It is worth to discuss the selection of specific machine learning method to develop an ensemble-based classifier.

Response 3: Thanks for the comment. In the revised manuscript the reason for adopting the ensemble-based classifier is explained in the last paragraph of section 3.2 System overview as:   

“The methods proposed previously encountered over-fitting issue when trained a single CNN model such as VGG16, VGG19 or ResNet models for larva classification. Therefore, we utilized the ensemble learning where multiple CNN models are trained individually were used as base models. The results of our experiment showed that the ensemble architecture achieves reliable performance than individual models.”

In addition, how the base models were selected is explained in the 4th paragraph of section 3.2.3. Base Model and Meta Model Selection as:

“For training the base models, we have investigated the transfer learning approach and evaluated 5 different CNN models: VGG16, VGG19, ResNet50, ResNet152 and In-ceptionV3 as candidates to find the best base models. The selection of candidate models was inspired by the previous research conducted in this field [21 –26, 28– 30 ].”

Point 4: Construction of test set should be briefed.

Response 4: The explanation of the text data is included in the last paragraph of section 3.2.2. Ensemble Learning for Larva Classification as

Another set of 100 ROIs unseen to the models was used for testing the proposed ensemble architecture in which it outperformed the existing methods in accuracy.”

Point 5: what is ? ? ? in page 372 and other places?

Response 5: the references are provided appropriately to fix this issue. 

Point 6: The data used for comparing other methods should be provided in the footnote.

Response 6: In section 4.2. Evaluation of Larva Classification the clarification of the data used for evaluating the other methods is given as: “However, the methods were evaluated using different datasets.”

Reviewer 2 Report

The article entitled “Aedes Larva Detection Using Ensemble Learning to Prevent Dengue Endemic” by Hossain et al points out the role of Vector-borne diseases, mainly Dengue among other deadly diseases such as Zika and Chikungunya caused by the parasites present in Aedes mosquito species and the need for cost effective identification method to control the spread of mosquitos. Here, the authors use machine learning, to detect Aedes larvae and distinguish from other mosquito species with an accuracy of 99% without needing any sophisticated microscopic magnifications but only at a max 4X magnification. Further, the authors describe in detail, the approaches to arrive at suitable base models and logistic regression (LR) as the meta classifier and further compared their proposed model with existing models in literature and prove the superior performance of the model. Overall, the use of these models, and less manual intervention would yield in early detection and control of these Vector-borne diseases and related deaths.

Major

  1. All images used for training and validation or at least a subset, and the Python/relevant scripts should all be made available on GitHub, and supplement a detailed documentation for reproducibility to the scientific community in this field. In case if the authors encounter restrictions/limitation over image size, then they should upload the scripts and README on GitHub, and consider providing an alternative site for the images to be accessible or provide the same as supplementary material
  2. The mention of base models in the manuscript is confusing “InceptionV3” at Line 323 (starting at Line 321), Line 370 and at Line 401. The paragraph from Line 401 to Line 404 should be moved and described in Results section, and at Line 323, mention not including base model " InceptionV3" (later described in results). This above statement is recommended; however, the authors can choose to correct this according to the feasibility.
  3. Any mention of name of organisms in the manuscript should follow ‘International Codes of Nomenclature’, please Genus and species name to italics and followed Sentence case. Example: Aedes aegypti.

Minor

  1. Expand ROC in ROC curves at Line 346
  2. ‘Tel:’ next to address of correspondence should be corrected or removed
  3. Line 34: Is it possible that the deaths could be related to COVID or other morbidities. Can the authors provide a reference supporting the statement “In 2019 the country experienced its largest outbreak with 101,354 cases and at least 266 deaths. The incident rate was alarmingly high in 2019 compared to the previous years.”
  4. Superscripts must be provided for all cases of series, at Line 87, for example, at Line 94 “4th instar”.
  5. Line 144: Remove full stop
  6. Line 190: Correct ‘comb’ vs ‘comp’
  7. Line 372, and Table 4 at Line 444, has missing references
  8. Only equation 2 and equation 4 are mentioned in manuscript. All, mention of equations should be made available in the text. Further, consistently address the numbers with in braces as shown in paragraph line 414.

Author Response

The article entitled “Aedes Larva Detection Using Ensemble Learning to Prevent Dengue Endemic” by Hossain et al points out the role of Vector-borne diseases, mainly Dengue among other deadly diseases such as Zika and Chikungunya caused by the parasites present in Aedes mosquito species and the need for cost effective identification method to control the spread of mosquitos. Here, the authors use machine learning, to detect Aedes larvae and distinguish from other mosquito species with an accuracy of 99% without needing any sophisticated microscopic magnifications but only at a max 4X magnification. Further, the authors describe in detail, the approaches to arrive at suitable base models and logistic regression (LR) as the meta classifier and further compared their proposed model with existing models in literature and prove the superior performance of the model. Overall, the use of these models, and less manual intervention would yield in early detection and control of these Vector-borne diseases and related deaths.

Major

Point 1: All images used for training and validation or at least a subset, and the Python/relevant scripts should all be made available on GitHub, and supplement a detailed documentation for reproducibility to the scientific community in this field. In case if the authors encounter restrictions/limitation over image size, then they should upload the scripts and README on GitHub, and consider providing an alternative site for the images to be accessible or provide the same as supplementary material

Response 1: Thanks for you very much for the comment. The Python scripts are uploaded on GitHub which is accessible though in the following link: https://github.com/Ezaz-Raihan/Aedes_Larva_Detection/blob/main/Ensemble_Stacking_in_Neural_Networks%20-%20Ezaz.ipynb

A subset of the dataset will be made available soon in the same location.

Point 2: The mention of base models in the manuscript is confusing “InceptionV3” at Line 323 (starting at Line 321), Line 370 and at Line 401. The paragraph from Line 401 to Line 404 should be moved and described in Results section, and at Line 323, mention not including base model " InceptionV3" (later described in results). This above statement is recommended; however, the authors can choose to correct this according to the feasibility.

Response 2: The manuscript is revised to clarify the why the InceptionV3 candidate was rejected as base model. The following lines are revised as:

2nd Paragraph of section 3.2.2. Ensemble Learning for Larva Classification:

“In this study, we trained 5 candidate models individually to predict the class label
for a given image which can be given as: fVGG16 : Iimage → C; fVGG19 : Iimage → C;
fResNet50 : Iimage → C; fResNet152 : Iimage → C; and fInceptionV3 : Iimage → C. After that, four
suitable models were selected among the candidates based on their performance. The
process of selecting the base models is explained in section 3.2.3. We have selected VGG16,
VGG19, ResNet50 and ResNet152 as based models and rejected InceptionV3. Then a logistic
regression classifier is used as the meta-model which is trained on the features derived
from the prediction scores of base models. The outputs of the base models were used in
the logistic regression model as the independent variable to predict the final class label.”

6th paragraph of section 3.2.3. Base Model and Meta Model Selection:

“From Table 3 it can be observed that VGG16, VGG19, ResNet50 and ResNet152 perform better than the InceptionV3. Therefore, we have selected the VGG16, VGG19, ResNet50 and ResNet152 models as the base model. Then, these models were stacked to construct the ensemble network to combat the over-fitting issue.”

Point 3: Any mention of name of organisms in the manuscript should follow ‘International Codes of Nomenclature’, please Genus and species name to italics and followed Sentence case. Example: Aedes aegypti.

Response 3: Thanks for the comment. The names of the organisms are corrected and written according to the given guideline.

Minor

Point 1: Expand ROC in ROC curves at Line 346

Response 1: ROC is expanded as receiver operating characteristics.   

Point 2: ‘Tel:’ next to address of correspondence should be corrected or removed

Response 2: It has been removed.

Point 3: Line 34: Is it possible that the deaths could be related to COVID or other morbidities. Can the authors provide a reference supporting the statement “In 2019 the country experienced its largest outbreak with 101,354 cases and at least 266 deaths. The incident rate was alarmingly high in 2019 compared to the previous years.”

Response 3: Thanks for the comment. Supporting reference has been given:

“In 2019 the country experienced its largest outbreak with 101,354 cases and at least 266 deaths \cite{ahsan2021possible}.”

Point 4: Superscripts must be provided for all cases of series, at Line 87, for example, at Line 94 “4thinstar”.

Response 4: Superscripts are provided for all the cases.

Point 5: Line 144: Remove full stop

Response 5: Full stop has been replaced with comma.  

Point 6: Line 190: Correct ‘comb’ vs ‘comp’

Response 6: Comp has been corrected as comb. 

Point 7: Line 372, and Table 4 at Line 444, has missing references

Response 7: The references are provided appropriately. 

Point 8: Only equation 2 and equation 4 are mentioned in manuscript. All, mention of equations should be made available in the text. Further, consistently address the numbers within braces as shown in paragraph line 414.

Response 8: All the equations are label and mentioned appropriately in the revised manuscript.